# Adverse Events of Percutaneous Microaxial Left Ventricular Assist Devices—A Retrospective, Single-Centre Cohort Study

**DOI:** 10.3390/jcm10163710

**Published:** 2021-08-20

**Authors:** Anna S. Zaiser, Gregor Fahrni, Alexa Hollinger, Demian T. Knobel, Yann Bovey, Núria M. Zellweger, Andreas Buser, David Santer, Hans Pargger, Caroline E. Gebhard, Martin Siegemund

**Affiliations:** 1Intensive Care Unit, University Hospital Basel, 4031 Basel, Switzerland; annaseraina.zaiser@usb.ch (A.S.Z.); alexa.hollinger@usb.ch (A.H.); demian.knobel@kssg.ch (D.T.K.); yann.bovey@ksa.ch (Y.B.); nuria.zellweger@usb.ch (N.M.Z.); hans.pargger@usb.ch (H.P.); martin.siegemund@usb.ch (M.S.); 2Department of Cardiology, University Hospital Basel, 4031 Basel, Switzerland; gregor.fahrni@usb.ch; 3Department of Clinical Research, University of Basel, 4031 Basel, Switzerland; 4Regional Blood Transfusion Center SRK Basel and Department of Hematology, Transfusion Medicine, University Hospital Basel, 4031 Basel, Switzerland; andreas.buser@usb.ch; 5Department of Cardiac Surgery, University Hospital Basel, 4031 Basel, Switzerland; david.santer@usb.ch

**Keywords:** Impella^®^, cardiogenic shock, acute heart failure, mechanical circulatory support

## Abstract

Worldwide, the left ventricular assist device Impella^®^ (Abiomed, Danvers, MA, USA) is increasingly implanted in patients with acute cardiogenic shock or undergoing high-risk cardiac interventions. Despite its long history of use, few studies have assessed its safety and possible complications associated with its use. All patients treated with a left-sided Impella^®^ device at the University Hospital of Basel from 1 January 2011 to 31 December 2019 were enrolled. The primary endpoint was the composite rate of mortality and adverse events (bleeding, acute kidney injury, and limb ischemia). Out of 281 included patients, at least one adverse event was present in 262 patients (93%). Rates of in-hospital, 90-day, and one-year mortality were 48%, 47%, and 50%, respectively. BARC type 3 bleeding (62%) and hemolysis (41.6%) were the most common complications. AKI was observed in 50% of all patients. Renal replacement therapy was required in 97 (35%) of all patients. Limb ischemia occurred in 13% of cases. Bleeding and hemolysis are common Impella^®^-associated complications. Additionally, we found a high rate of AKI. A careful selection of patients receiving microaxial LV support and defining the indication for its use are essential measures to be taken for the benefits to outweigh potential complications.

## 1. Introduction

Cardiogenic shock (CS) is a state of decreased cardiac output, defined as systolic blood pressure below 90 mmHg with signs of hypoperfusion, despite an adequate filling state [1]. Reduced cardiac output in the presence of a fully loaded ventricle, therefore, increases left ventricular end-diastolic volume, pressure, wall tension, and, thus, myocardial oxygen consumption. This leads to a vicious circle of decreased coronary perfusion, which further reduces contractility and continues to worsen ventricular function [2]. One way to counteract this downward spiral is by unloading the ventricle through a mechanical assist device such as the Impella^®^ (Abiomed, Danvers, MA, USA). The Impella^®^ is a microaxial, minimally invasive ventricular assist device, based on the principle of an Archimedes screw. Depending on the desired flow rate, there are five different types of Impella^®^ pumps for insertion into the left ventricle (LV): the Impella 2.5^®^, CP^®^, LD^®^, 5.0^®^, and 5.5^®^ (the last three requiring surgical cut-down).

Impella^®^ devices are used to unload the left or right ventricle and to simultaneously improve systemic perfusion as well as increase coronary blood flow [3]. These devices thereby help to reduce LV stroke effort and as a result myocardial oxygen consumption. This is especially important in CS or during high-risk percutaneous coronary interventions (PCIs) [4,5]. The introduction of the Impella^®^ has led to a paramount change in mechanical circulatory support (MCS) strategies. For specific indications it has replaced other MCS devices including the intraaortic balloon pump (IABP) [6] and the extracorporeal membrane oxygenation (ECMO) [7]. However, the only large prospective, randomised clinical trial comparing the use of the IABP and the Impella^®^ in patients undergoing high-risk PCIs between 2007 and 2010, the PROTECT II study, was terminated after a first interim analysis because of suspected futility [8]. A comparable prospective randomised clinical trial for the treatment of CS with and without PCI is lacking and outcome data in patients presenting with CS are sparse. Although the concept of unloading the LV in CS is appealing, only two small trials comparing the Impella^®^ to the IABP in CS have been published [9,10]. Both have shown no improvement in survival and a higher incidence of complications with percutaneous MCS. Nonetheless, some retrospective studies have investigated the safety and possible complications of the Impella^®^ pump [11,12,13,14]. Increased use of this microaxial pump calls for more detailed assessments of adverse effects and therapeutic options. The aim of this study was to describe and analyse a single-centre experience concerning mortality and adverse events of patients using an Impella^®^ percutaneous microaxial LV assist device.

## 2. Materials and Methods

Data, analytic methods, and study materials will be available to other researchers upon request. Th study was approved by the Ethics Committee of Northwestern and Central Switzerland (EKNZ, Project ID 2021-00002). According to Swiss law, data could be collected if no statement against the elicitation of quality data was existent.

### 2.1. Study Population

This observational, retrospective, single-centre analysis investigated all patients who received LV Impella^®^ support from January 2011 until December 2019 at the University Hospital of Basel (USB), irrespective of indication for insertion. Patients in whom an Impella^®^ was implanted more than once were assessed as separate cases for each application. Resuscitation before insertion of the Impella^®^ was defined as any mechanical or electric cardiopulmonary resuscitation that did not take place during a cardiac intervention or surgery.

For all patients without available mortality data, the date of the last blood draw in our hospital was assessed. If it was longer than one year after the Impella^®^ insertion, the patient was considered still alive. The patient was also considered to be alive if the last blood draw was within one year after the Impella^®^ insertion and if the patient still had a registered social security number. If the latter was not the case, data were declared unavailable, but the patient was not declared dead.

To compare our results to the recent study from the Acute Myocardial Infarction in Switzerland (AMIS) Plus Registry regarding the trend associated with the outcome of cardiogenic shock [15], we performed a subgroup analysis of all patients where a CS with acute myocardial infarction (AMI) was the reason for Impella^®^ insertion.

According to standard protocol, anticoagulation was initiated with 5000 IU of heparin just before implantation, and maintained with therapeutic heparin levels (target anti-factor Xa activity of > 0.26 U/mL).

### 2.2. Data Collection

Data including patient demographics, medical history, laboratory data, procedural details, in-hospital adverse events, and clinical outcomes including mortality were collected from the patient’s medical record. Data from patients who received an Impella^®^ device at the University Hospital and who were later transferred to another hospital are included with Basel data only, except mortality. Data on the indication for Impella^®^ support were collected from the cardiology report. The type of Impella^®^ supporting the patient for the longest period of time was registered. Change of Impella^®^ was counted as a new case only when another type was used. When the same Impella^®^ was reinserted into the contralateral limb, it was counted as the same case.

### 2.3. Definition of Adverse Events and Outcomes

Data on adverse events were collected during Impella^®^ placement, hospital stay, and until one year after discharge.

Limb ischemia after Impella^®^ placement was declared when signs of ischemia (i.e., pain, pallor, paresthesia, poikilothermia, pulselessness, and/or paralysis) were mentioned in the medical records at the site of the microaxial pump insertion for more than one day or if surgical intervention (i.e., shunt placement, bypass surgery, and/or thrombendarterectomy) was documented.

The patients were counted as an acute stroke if a new neurological deficit with signs of malperfusion or bleeding in cranial imaging occurred during their hospital stay or was reported in the patient’s records within one year. If medical records later than one year after Impella^®^ insertion documented no stroke so far, the patient was counted in the “no stroke” group. If there was no report at all, we placed the patient into the group “data not available”. All peripheral thrombotic complications were also registered.

Hemolysis was defined as a change in bilirubin of ≥ 10 µmol/l accompanied by a concurrent increase in lactate dehydrogenase (LDH) of ≥ 558 U/l from the last value before Impella^®^ implantation to the peak value during Impella^®^ support, or if haptoglobine was below 0.3 g/L.

Bleeding was classified according to the Bleeding Academic Research Consortium (BARC) criteria [16]. Access site bleeding was classified as BARC types 1 (minor) and 2 (more severe). All bleeding requiring surgical exploration or transfusion of blood products during intensive care unit (ICU) stay was classified as BARC type 3. Moreover, all cardiac tamponades, hemorrhagic shock without death, and intracranial hemorrhages were also classified as type 3. All patients who received cardiac surgery and required transfusion were classified as BARC type 4. Patients who developed hemorrhagic shock and died in hospital were classified as BARC type 5. Hemorrhagic shock associated with cardiac surgery was analysed separately. The number of red blood cell (RBC) transfusions (units) given in the ICU or in the operating room (OR) were only counted during the time of Impella^®^ therapy.

Thrombocytopenia was defined as platelets below 100 × 10^9^/L, or, if already below 100 × 10^9^/L at the time of Impella^®^ insertion, the decrease of > 50% from baseline during the time of ventricular mechanical support was counted as Impella^®^-associated thrombocytopenia. If no platelet measurement was available during Impella^®^ support, these patients were excluded from the analysis of thrombocytopenia.

Acute renal dysfunction was defined according to the “Kidney Disease: Improving Global Outcomes” (KDIGO) stages [17]. Our baseline value was defined as the last creatinine value before Impella^®^ insertion. The observational period included seven days after the start of LV mechanical support. Acute liver injury during Impella^®^ support was defined as a rise in alanine aminotransferase (ALAT) of > 1000 U/L (17 times above upper laboratory limit) during ICU stay. Multiple organ failure was defined as simultaneous failure of two or more organs, heart included, during the ICU stay. Worsening of valve function during Impella^®^ support was assessed for the mitral valve and the aortic valves separately. Cardiac tamponade was registered regardless of a causal correlation with the Impella^®^. LV ejection fraction (LVEF) before Impella^®^ insertion (maximum 3 months before insertion) and without mechanical circulatory support either at discharge or before death was collected from echocardiographic reports or daily records. In case LVEF values were available only before Impella^®^ insertion, these values were also used to calculate LVEF at discharge/last before death. Descriptive statements concerning the severity of the LV dysfunction, and not two-dimensionally calculated, were defined as follows: mild dysfunction at LVEF of 40%, moderate dysfunction at LVEF of 30%, and severe dysfunction at LVEF of 20%. When LVEFs of < 30%, < 20%, or < 10% were listed on reports, we designated these as LVEFs of 25%, 15%, or 10%, respectively. Disease severity was assessed according to the Simplified Acute Physiology Score (SAPS) II score at the time of the ICU admission.

Dual antiplatelet therapy (DAPT) was defined as prescription of aspirin and clopidogrel, ticagrelor, or prasugrel simultaneously. Administration of ≥ 10′000 IE unfractionated heparin per day was counted as therapeutic heparinisation. 

### 2.4. Study Endpoints

The primary endpoint was the rate of adverse events. An adverse event was defined as death, stroke, ischemic complications (non-central nervous system), vascular complications (aneurysms, dissections, or arteriovenous fistulas), bleeding (BARC types 3–5), novel posterior mitral valve chordal rupture, acute kidney injury (AKI), need for renal replacement therapy (RRT), limb ischemia, amputation, and multiorgan failure.

Secondary endpoints were length of ICU stay, duration of Impella^®^ support, LVEF at discharge or last before death, bleeding (BARC types 1–2), hemolysis, and thrombocytopenia.

### 2.5. Statistical Analysis

All categorical variables are expressed in percentages. Continuous values are presented as medians with interquartile ranges (IQRs). In cases of early death, values concerning thrombocytopenia and AKI are incomplete for a small number of patients. Here we performed two analyses to show that the number of the cases with missing data did not have an impact on the results. At the University Hospital of Basel, patients presenting with acute cardiogenic shock are being treated routinely with an Impella^®^; the IABP is not being used. Therefore, we did not perform a propensity score matching or matching of any other kind, because our cohort does not have a comparable number of cardiogenic shock patients treated conservatively.

## 3. Results

Over the nine-year studied period (2011–2019), a total of 281 patients received LV mechanical support with the Impella^®^ (Table 1). The most common indication for Impella^®^ insertion was ischemic CS (70%). One patient received an Impella^®^ twice within a few days. The median SAPS II score was 53 (IQR, 37–68) in survivors and 70 (IQR, 54–83) in non-survivors. The SAPS II score was higher in patients who died during their hospitalisation than in those who died after hospital discharge (73 (IQR, 60–85) and 54 (IQR, 42.3–72.3), respectively). Median duration of Impella^®^ support was 72 h (IQR, 35–121). The maximum duration in one patient was 493 h (20.5 days). A total of 262 patients (93%), experienced an adverse event. With the exclusion of patients without available data on stroke and mortality, 250 out of 269 (93%) experienced an adverse event. Bleeding was the most frequent adverse event, as shown in Table 2 and Figure 1. After excluding bleeding complications, a total of 261 out of 281 patients (93%) suffered an adverse event. After excluding patients without available data on stroke and mortality, 249 out of 256 (96%) suffered an adverse event. Complications and outcome parameters are described in Table 2. Details on adverse events stratified by sex (Appendix A) and age categories (Appendix A) are shown in the Appendix A. Almost half (48%) of all patients died during hospitalisation and overall mortality was very high with a total of 165 (59%) deceased patients. Of the patients, 43 (15%) died within the first 24 h after Impella^®^ placement, while a total of 95 (34%) patients died within one week. Thirty-day mortality was 46%, while 50.5% of patients had died after one year, as shown in Table 2 and Figure 2.

The most frequent complication was BARC type 3 bleeding (62%), followed by BARC types 1 and 2 (37%). The maximum number of RBC units transfused to one surgical patient was 81. Details of organ failure assessment are described in Table 3. Fifty percent of all patients developed an AKI (KDIGO stage 1–3), and 23% developed acute kidney failure (KDIGO stage 3). In patients with KDIGO stage 3, 64% required RRT. Fifty-six percent of patients developed hemolysis during Impella^®^ support. Thrombocytopenia was detected in 44% of patients. In patients suffering from bleeding, the median lowest platelet count was 74 x 109/l (IQR, 44.3–131), as shown in Table 4.

## 4. Discussion

In this retrospective single-centre study we described 281 patients undergoing LV MCS with an Impella^®^ device. To our knowledge, this is the largest cohort to describe the most common complications of Impella^®^ support. First, 93% of all patients developed an adverse event, although we could not always attribute the cause distinctively either to CS or the Impella^®^. Second, bleeding was the most common complication (71% of all patients), followed by hemolysis (in 41.6%). Third, in-hospital (47.7%) and one-year mortality (50.5%) was comparable to other CS cohorts [14,15]. Fourth, the rate of AKI was at 50% in our patient cohort, with RRT being required in 35%. Fifth, acute myocardial infarction was the reason for Impella^®^ implantation in more than two thirds of the patients (70%).

Our mortality rate is comparable to previously reported rates in the Impella mechanical circulatory support device in Italy (IMP-IT) registry [14]. A potential explanation for these rates may be the severity of illness and an already increased risk of death in patients requiring LV support via a microaxial pump. Complicating factors such as bleeding and hemolysis are in themselves also associated with increased mortality [18,19,20]. However, we were not able to show a reduction in mortality compared to Swiss AMI patients with CS in the AMIS Plus Registry [15]. Their reported in-hospital mortality of 49.2% represents the overall mortality in AMI patients with CS between 1997 and 2017. The authors reported a decrease in mortality from 47.3% (2007–2013) to 41.0% (2014–2017) over the entire study period. This may indicate that the use of a microaxial pump in CS might not reduce mortality as presented in previous studies, although our cohort only included very severe CS patients [21,22,23,24]. Therefore, the documented reduction of the mortality over time in the AMIS Plus Registry might not be correlated with the increased use and benefit of the Impella^®^.

In the existing literature, hemolysis is described as having a rate of 10–62.5% under Impella^®^ support. Badiye et al. [25] reported a hemolysis rate of 62.5% and suggested a higher incidence of hemolysis in patients with prolonged LV mechanical support. Another study reported malpositioning of the microaxial pump as the most common cause for their hemolysis incidence of 18.6% [20], but did not give a physiologic explanation for this correlation. This was also pointed out in a study from Burzotta et al. [3], though they only reported a rate of hemolysis rate of < 10% during long-term LV mechanical support. In their review, Subramaniam et al. [26] found an occurrence of hemolysis between 10–46%. They also postulated that longer duration of microaxial pump support to be related to ongoing hemolysis. Furthermore, hemolysis is often associated with the design of the device, being of small size and providing a microaxial flow [20,27,28]. This could cause the destruction of particularly older erythrocytes with less flexible cell walls, mostly likely occurring at the start of treatment using a microaxial pump [29]. In addition, the wide variation of clinical cut-offs defining hemolysis in previous studies may also contribute to the differences in the reported rates.

In our study, hemolysis was detected in 41.6% of all patients. In 71 out of 281 patients their values concerning hemolysis were missing. It was therefore assumed that these patients did not suffer from hemolysis. As the Impella^®^ has been in use for nearly two decades in our hospital and appropriate positioning was regularly confirmed by thoracic radiography or cardiac echocardiography, experience of proper placement can be assumed to be high. Furthermore, the highest LDH plasma concentrations could be measured early after the insertion of the microaxial pump. Therefore, malposition seems to be less likely to be responsible for the high rate of hemolysis observed in our cohort. As previously stated, there is a need for consensus in cut-offs to define hemolysis and to improve comparability across studies [28]. Our suggestion is to measure haptoglobin during the Impella^®^ therapy to unveil concomitant hemolysis, as severe long-lasting hemolysis may increase the rate of AKI in these patients. Exceptionally high LDH levels may foster the decision to withdraw the device.

In a review published in 2019 [26], a wide range of major bleeding from 0 to 54% was reported. Several studies described a correlation between the use of the microaxial pump and an increased risk of bleeding [12,23]. Bleeding was the most common complication in our study. BARC type 3 bleeding occurred in 62% of all patients, whereas BARC types 1 and 2 (access site bleeding) were seen in 37%. Comparison of our subgroup of CS patients with AMI to the AMI Switzerland cohort showed a difference in bleeding from 60% to 7%. We matched BARC type 3 bleeding in our cohort to the AMIS cohort, although they did not specify grades of bleeding. Compared to our cohort, only 10% of the AMIS patients were supported by either an ECMO or microaxial pump [15]. Thus, the main reason for this difference in bleeding might be related to the indication for LV device implantation.

One reason for the high incidence of bleeding could be the larger bore access site needed for the Impella^®^ compared to the IABP, as previously described [11,12,30]. Other reasons could be the need for continuous anticoagulation in the presence of dual platelet inhibition during LV mechanical support [26] or the higher incidence of coagulopathic disorders in patients with multiple organ failure [31]. In our cohort only 53% of bleeding patients were treated with dual platelet inhibition. Furthermore, previous studies have shown a correlation between short-term MCS and the development of acquired von Willebrand syndrome [32,33,34]. This is due to the destruction of the large molecular precursor protein ADAMTS-13, and may have contributed to the high occurrence of bleeding in our study. However, as parameters indicating acquired von Willebrand syndrome had not been recorded, we have no conclusive evidence about the incidence of acquired von Willebrand syndrome.

AKI occurred in 50% of our overall cohort, and RRT was required in 35% of all patients. The reported rates of AKI (1% to 32%) during the use of the microaxial pump vary widely [8,21,22,24,35,36,37,38], as does the rate of patients receiving RRT, with reported rates ranging from 14% to 42% [10,28,38,39,40]. Amin et al. showed an increase in AKI from the pre-Impella^®^ to the Impella^®^ era with an odds ratio of 1.91 [11]. A meta-analysis from 2018 demonstrated a higher risk of AKI associated with the use of the Impella^®^ [41]. Due to the retrospective nature of our analysis, we are not able to differentiate between AKI caused by the LV mechanical support or the mere coincidence of the implanted pump and AKI due to CS, as acute CS itself is a risk factor for developing AKI due to end-organ hypoperfusion and congestion. In fact, renal failure is often used as one of the diagnostic criteria of CS [2]. Vallabhajosyula et al. [42] and Tarvasmäki et al. [43] have reported rates of AKI in patients with CS of 35% and 31%, respectively. This may suggest that support from the Impella^®^ might increase the incidence of AKI in CS patients because of the simultaneous occurrence of CS, hemolysis, and the increased need for red blood cell transfusion [44].

Limb ischemia occurred in 13% of all cases, but no patients needed an amputation. The Italian IMP-IT registry showed an 13% incidence of limb ischemia in patients with CS [14]. For patients undergoing high-risk PCI with the use of this microaxial device, limb ischemia was reported in only 2.8% of patients. Other studies showed an incidence between 1–4% [3,21,37,40,45,46]. A recent review [26] found an occurrence of limb ischemia between 0.1–10% and recommended careful and continuous monitoring of distal perfusion during Impella^®^ therapy. O’Neill et al. [21] suggest that an elective antegrade perfusion cannula might be useful to reduce acute limb ischemia, especially when hemodynamic instability requires prolonged mechanical circulatory support.

Despite the concerning number of adverse events, we think that careful insertion using vascular ultrasound to avoid direct device complications will be beneficial in patients with CS. Similarly, a proper selection of patients and the shortest possible percutaneous LV assist may improve the risk–benefit ratio.

### Limitations

This study has several limitations. First, it was performed as a retrospective single-centre study, and therefore may not be representative for the general population. Second, data collection was retrospective and is, therefore, subject to recall and ascertainment bias as well as being unable to provide answers of causality. Third, a control group for the comparison of patients in acute cardiogenic shock is missing given the predominant treatment regimen of severe cardiogenic shock patients at our institution. In addition, a multivariate analysis is not provided. Therefore, our findings remain only hypothesis generating. Forth, data on adverse events after hospital discharge are not provided, since readmission and patient follow-up was not necessarily performed at our institution. Fifth, the specific type of Impella^®^ inserted was not always recorded; and its rotational speed was not assessed. This may be important information to determine the origin and incidence of hemolysis. However, these limitations may be used to plan and conduct further prospective and randomised studies in this field.

## 5. Conclusions

In our study, 34% of patients in CS and Impella^®^ assistance died within the first week, while one-year mortality was 50.5%. At the same time, we found a relatively high rate of severe bleeding, hemolysis, and AKI. These outcomes are in-line with previous studies showing a mortality in the same range and an increased rate of complications. In particular, the relatively higher rate of AKI compared to other cohorts with AMI-related CS underlines the urgent need for a prospective, randomised multi-centre study to elucidate the value of microaxial, minimally invasive ventricular assist devices in CS. Given the limitations of our study design, our findings should be considered hypothesis generating and serve as a catalyst to prospectively assess the safety, indication, and potential complications of microaxial, minimally invasive LV assist devices.

## Figures and Tables

**Figure 1 jcm-10-03710-f001:**
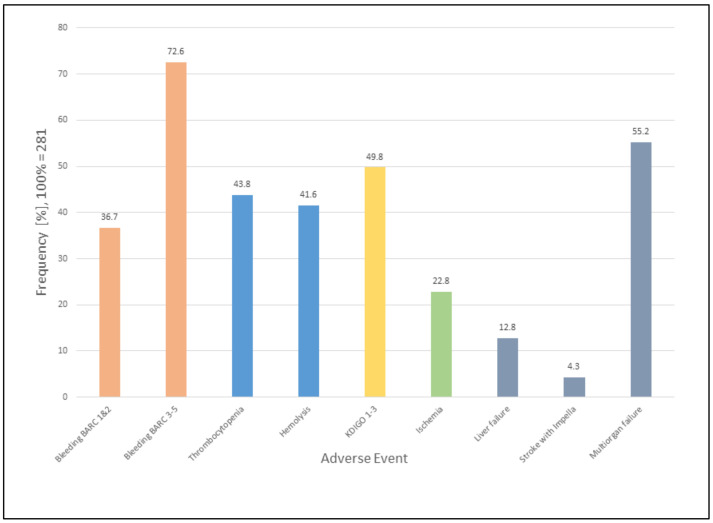
Frequency of adverse events. Numbers are percentages. For hemolysis, 71 out of 281 values were missing. It was assumed that these patients did not suffer from hemolysis.

**Figure 2 jcm-10-03710-f002:**
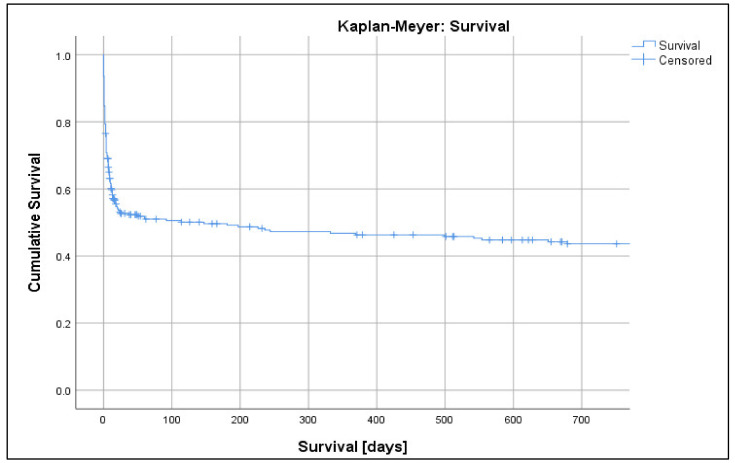
Kaplan–Meyer survival analysis.

**Table 1 jcm-10-03710-t001:** Patient characteristics, procedures, and device information.

Characteristics, *n* = 281	*n* (%), Median (IQR)
Age, years	65 (57–75)
Male	210 (74.7)
Length of ICU stay, days	7 (4–11)
Length of stay of patients that left ICU alive, days	8 (5–13.5)
SAPS II score	63.5 (47–77)
Resuscitation before Impella^®^ insertion	113 (40.2)
Immediate cardiac surgery prior to Impella^®^ insertion	25 (8.9)
CABG only	13 (52)
Valve surgery	5 (20)
Other	7 (28)
Urgent (24 h), emergency (12 h), and salvage surgery	15 (60)
Elective surgery	10 (40)
**Anticoagulation and Antiplatelet Treatment**	
DAPT	171 (60.9)
In patients with bleeding complications (*n* = 198)	104 (52.5)
In patients without bleeding complications (*n* = 83)	67 (80.7)
Heparin, unfractionated and therapeutic	244 (86.8)
In patients with bleeding complications (*n* = 198)	175 (88.4)
In patients without bleeding complications (*n* = 83)	69 (83.1)
aPTT (*n* = 231)	82 (54–181)
In patients with bleeding complications	95 (58–181)
In patients without bleeding complications	64 (46–106)
INR (*n* = 280)	1.35 (1.2–1.8)
In patients with bleeding complications	1.4 (1.2–1.9)
In patients without bleeding complications	1.2 (1.1–1.5)
**Indication for Impella^®^ Use**	
CS acute myocardial infarction	196 (69.8)
CS ischemic without acute myocardial infarction	8 (2.9)
CS valvular	18 (6.4)
CS cardiomyopathy	19 (6.8)
CS other diagnosis (rhythmogenic, myocarditis)	34 (12.1)
No CS (after ECMO removal, surgery under Impella^®^ support, etc.)	6 (2.1)
Type of Impella^®^	
Impella 2.5^®^	143 (50.9)
Impella CP^®^	123 (43.8)
Impella 5.0^®^	6 (2.1)
Not available	9 (3.2)
Additional Impella^®^ RP	4 (1.4)
Change of Impella^®^ to another type	8 (2.9)
Duration of Impella^®^ support, h (median, IQR)	72 (35–121)
Escalation therapy (ECMO, LVAD, or heart transplantation)	18 (6.4)
ECMO and additional Impella^®^ as LV vent	9 (3.2)
**Cardiac Function and Outcome Parameters**	
LVEF prior to Impella^®^ implantation	28 (20–40)
LVEF at discharge/last before death	35 (25–43)
In-hospital death	25 (15–35)
Patients discharged from hospital	40 (32–45.5)

IQR, interquartile range; ICU, intensive care unit; SAPS, Simplified Acute Physiology Score; DAPT, dual antiplatelet therapy; aPTT, activated partial thromboplastin time; INR, international normalised ratio; CS, cardiogenic shock; ECMO, extracorporeal membrane oxygenation; LVAD, left ventricular assist device; LV, left ventricle; and LVEF, left ventricular ejection fraction.

**Table 2 jcm-10-03710-t002:** Adverse events.

Mortality Data, *n* = 281	*n* (%), Median (IQR)
Survived	91 (32.4)
Deaths	165 (58.7)
Not available	25 (8.9)
In hospital	134 (47.7)
1-day mortality	43 (15.3)
3-day mortality	66 (23.5)
7-day mortality	95 (33.8)
30-day mortality	129 (45.9)
90-day mortality	133 (47.3)
1-year mortality	142 (50.5)
**Adverse Events**	
Bleeding *	
No bleeding	83 (29.5)
BARC types 1 and 2	103 (36.7)
BARC type 3	175 (62.3)
BARC type 4	24 (8.5)
BARC type 5	5 (1.8)
Number of all patients needing transfusions ^†^	163 (58)
Number of RBC transfusions, units (*n* = 163, median (IQR))	4 (2–12)
Stroke within one year	21 (7.5)
Stroke during Impella^®^ therapy	12 (4.3)
Ischemic complications (non-central nervous system) ^‡^	56 (19.9)
Intestinal ischemia	25 (8.9)
Limb ischemia	36 (12.8)
Thrombotic complications (jugular venous)	3 (1.1)
Worsening of the valve function^‡^	22 (7.8)
Aortic valve	6 (2.1)
Mitral valve	14 (5)
New posterior mitral valve chordal rupture	4 (1.4)
Hemorrhagic shock	9 (3.2)
Associated with cardiac surgery (included in BARC type 3 bleeding)	4 (1.4)
All vascular complications (aneurysms, dissections, or arteriovenous fistulas)	12 (4.3)
Vascular complications without surgery	4 (1.4)

BARC, Bleeding Academic Research Consortium; RBC, red blood cells; IQR, interquartile range; and LVEF, left ventricular ejection fraction. *, During the ICU stay, multiple types possible. †, During Impella^®^ therapy. ‡, Multiple factors possible.

**Table 3 jcm-10-03710-t003:** Organ failure.

Organ Failure Assessment	*n* (%)
Pre-existing chronic kidney disease (not CS related)	40 (14.2)
KDIGO Stage with missing values	
0	141 (50.2)
1	38 (13.5)
2	38 (13.5)
3 *	64 (22.8)
KDIGO Stage 3 + RRT with missing values (*n* = 64)	41 (64.1 ^†^)
KDIGO Stage without missing values (*n* = 273)	273 (97.2)
0	141 (51.7 ^†^)
1	38 (13.9 ^†^)
2	38 (13.9 ^†^)
3	56 (20.5 ^†^)
KDIGO Stage 3 + RRT without missing values (*n* = 56)	39 (69.6 ^†^)
KDIGO Stage for patients with hemolysis (*n* = 117)	
0	40 (34.2 ^†^)
1	22 (18.8 ^†^)
2	21 (18 ^†^)
3	34 (29.1 ^†^)
KDIGO Stage for patients without hemolysis (*n* = 93)	
0	61 (65.6 ^†^)
1	8 (8.6 ^†^)
2	10 (10.8 ^†^)
3	13 (14 ^†^)
RRT received	97 (34.5)
Regular dialysis prior to hospitalisation	3 (1.1)
RRT and death (*n* = 97)	97 (34.5)
In hospital	64 (66^†^)
After hospitalisation	9 (9.3 ^†^)
Liver failure	36 (12.8)
Multiorgan failure	
two organs	67 (23.8)
≥ three organs	88 (31.3)

CS, cardiogenic shock; KDIGO: Kidney Disease: Improving Global Outcomes; and RRT, renal replacement therapy. *, All eight patients without a second creatinine measurement due to early death were included in this subgroup. †, Percentage of the corresponding subgroup.

**Table 4 jcm-10-03710-t004:** Hemolysis and thrombocytopenia.

Hemolysis and Thrombocytopenia	*n* (%), Median (IQR)
Hemolysis	117 (41.6)
Hemolysis without missing values (*n* = 210)	117 (55.7)
Thrombocytopenia	123 (43.8)
Nadir of platelets, 10^9^/l	
With missing values (*n* = 279)	108 (60.5–158.5)
Without missing values (*n* = 271)	104 (60–157)
Nadir of platelets for patients with thrombocytopenia, 10^9^/l	
With missing values (*n* = 131)	62 (36.5–78)
Without missing values (*n* = 123)	60 (34.5–76)
Nadir of platelets & bleeding, 10^9^/l	
With missing values (*n* = 170)	74 (42.75–131)
Without missing values (*n* = 168)	74 (44.25–131)
Nadir of platelets and no bleeding, 10^9^/l	
With missing values (*n* = 109)	146 (103–203)
Without missing values (*n* = 103)	143 (103–199)

## Data Availability

The data presented in this study are available on request from the corresponding author. The data are not publicly available due to privacy and ethical reasons.

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
