# Peer review of "Adverse Events of Percutaneous Microaxial Left Ventricular Assist Devices—A Retrospective, Single-Centre Cohort Study"

_jcm, 2021, doi:10.3390/jcm10163710_

Round 1

Reviewer 1 Report

This review reports on a single-centre experience concerning mortality and adverse events of patients presenting with cariogenic shock and using an Impella® percutaneous microaxial LV assist device. 

The paper is well written, informative and refers to pertinent publications in the field. However, it could certainly be improved from a more structured comparison of this group treated with Impella against alternative procedures (for instance medical therapy or IABP) in order to understand which is the standard of care in your Center (mortality, adverse events, etc) and the severity of patients. This is a main limitation of the study .

Other concerns:

1- The paper does not provide any information regarding associated procedures (PTCA or CABG) and medical therapy (double antiplatelet therapy, INR). It may be useful to better understand the causes of in-hospital mortality.

2- An univariable and multivariable analysis is missing.

3- There are only early results and no follow-up of patients treated. One would expect to see at follow-up data on survival, freedom from cardiac, neurological or  vascular events etc. (after discharge).

4- The Discussion is too long and it should be concentrated in commenting the authors results in comparison with those present in the current literature and not just repeat the already reported results point by point.

5- Which are the final implications of your results? A final message is not clear and the conclusions should to be reduced and concentrated. 

Author Response

Answers to Reviewer Comments

Manuscript jcm-1288047: Adverse Events of Percutaneous Microaxial Left Ventricular Assist Devices - A Retrospective, Single-Centre, Cohort Study

 We would like to thank all reviewers for their most constructive comments, which are addressed in detail below in a point-by-point rebuttal. The resulting insights have been included in the revised version of the manuscript (highlighted in yellow colour).

Reviewers' comments:

Reviewer #1:

This review reports on a single-centre experience concerning mortality and adverse events of patients presenting with cardiogenic shock and using an Impella® percutaneous microaxial LV assist device. 

The paper is well written, informative and refers to pertinent publications in the field. However, it could certainly be improved from a more structured comparison of this group treated with Impella against alternative procedures (for instance medical therapy or IABP) in order to understand which is the standard of care in your Center (mortality, adverse events, etc) and the severity of patients. This is a main limitation of the study.

We thank the reviewer for his general favorable comment. We would like to have the opportunity to compare our Impella patients with a cohort of patients with IABP or sophisticated medical therapy. Unfortunately, our cardiology department did not use any IABP since more than 15 years and thus, comparison with a group of patients with counter-pulsation was not possible. In addition, almost every patient with severe CS in the cardiology Cath Lab receives an Impella pump, while only view patients remain with medical therapy, preventing a useful propensity match, because of the different severities of CS. We stated this in the statistical analysis section (page 8/9, lines 192-196) and added it to the limitation section (page 13, lines 337-339).

Comment #1.1: The paper does not provide any information regarding associated procedures (PTCA or CABG) and medical therapy (double antiplatelet therapy, INR). It may be useful to better understand the causes of in-hospital mortality.

Response to comment #1.1: Thank you for the important remark. In our institution, Impella support is almost exclusively used in cardiogenic shock patients from acute myocardial infarction undergoing salvage PCI (69.8% of patients), while intraoperative insertion of single Impella assists during cardiac surgery is rare. Indeed, most patients undergoing cardiac surgery were either urgent or salvage coronary-artery bypass graftings (n= 15, 60%), combined surgeries (CABG and valve) or Impella was only used as LV vent in patients with ECMO devices (n=9, 3.2%).  Table 1 now provides more detailed information on cardiac interventions.

In addition, we now provide information on medical treatment (dual antiplatelet therapy) and therapeutic dose of unfractionated heparin in patients with bleeding complications. In patients with an Impella assist, therapeutic unfractionated heparin is routinely used for anticoagulation. This information is now provided in Table 1.

Comment # 1.2: An univariable and multivariable analysis is missing.

Response to comment #1.2: We agree with the reviewer, that a univariable or multivariable analysis would be beneficial. We performed a survival analysis of all patients in our cohort and included it into the manuscript, new Figure 2.   Given the very short revision dead-line and the holiday absence of our statistical adviser, we were not able to perform a conscientious and comprehensive multivariate analysis, which is now stated in the limitation section (page 13, line 339), further underlining the hypothesis generating character of this report.

Figure 2: Kaplan Meyer Survival Analysis

 Comment #1.3: There are only early results and no follow-up of patients treated. One would expect to see at follow-up data on survival, freedom from cardiac, neurological or  vascular events etc. (after discharge).

Response to comment 1.3: We agree with the reviewer that long-term follow-up data on freedom from adverse events would be beneficial. However, given the retrospective design of our study, the high mortality rate within 1 year after hospital discharge and patients not necessarily being readmitted to our institution in case of adverse events, we are not able to provide all data requested. In addition, we only assessed data on adverse events from the initial hospital admission with cardiogenic shock and Impella treatment and not from subsequent hospital admissions, as we think that no causal relation to former Impella treatment could exist. Nevertheless, data on mortality and neurologic outcomes were collected over more than one year. In terms of neurological outcomes, our institution is the only neurologic referral center in the commuting area of our University hospital, so that the number of missing neurologic events is small. Data on stroke within one year after hospital discharge are provided in Table 2. Mortality was evaluated with the aid of the central Swiss social security number (last data access on 22 July 2021). The missing information on adverse events is now mentioned in the limitation section, page 13/14, lines 340-342.

Comment #1.4: The discussion is too long and it should be concentrated in commenting the authors results in comparison with those present in the current literature and not just repeat the already reported results point by point.

Response to comment #1.4: We agree with the reviewer and adapted the discussion accordingly. The discussion was shortened (one page) and our findings are now discussed in the context of the existing literature. We highlighted the areas where we discarded sentences in the manuscript.

Comment #1.5: Which are the final implications of your results? A final message is not clear and the conclusions should to be reduced and concentrated. 

Response to comment #1.5: We thank the reviewer for this important comment. The conclusion was shortened and now focuses on a final message. Page 14, lines 348 -357.

Reviewer 2 Report

Zaiser et al. reported epidemiological features in a retrospective manner on adverse events after using the Impella device on cardiogenic shock, which has been overlooked. The authors have made significant progress towards establishment for its detailed indications and preventive measures for adverse events.

It would be better to address some concerns for scientific soundness before the acceptance of the manuscript.

  1. Most mortality occurred in hospitals, with 30 days and six months of death being a minority. It would be helpful to present the mortality rate in the hospital as the day, three days, and seven days.
  2. The number of patients analyzed appears to be 281, but in the first sentence of the Discussion, 282 patients are presented. I encourage you to provide an accurate number.
  3. If possible, it would be meaningful to describe whether there are differences in adverse events depending on age groups and gender.

Author Response

Answers to Reviewer Comments

Manuscript jcm-1288047: Adverse Events of Percutaneous Microaxial Left Ventricular Assist Devices - A Retrospective, Single-Centre, Cohort Study

 We would like to thank all reviewers for their most constructive comments, which are addressed in detail below in a point-by-point rebuttal. The resulting insights have been included in the revised version of the manuscript (highlighted in yellow colour).

Reviewers' comments:

Reviewer # 2:

Zaiser et al. reported epidemiological features in a retrospective manner on adverse events after using the Impella device on cardiogenic shock, which has been overlooked. The authors have made significant progress towards establishment for its detailed indications and preventive measures for adverse events. It would be better to address some concerns for scientific soundness before the acceptance of the manuscript.

Comment #2.1: Most mortality occurred in hospitals, with 30 days and six months of death being a minority. It would be helpful to present the mortality rate in the hospital as the day, three days, and seven days.

Response to comment #2.1: We thank the reviewer for this important comment. We now provide more detailed data on in hospital mortality, according to the reviewers’ suggestions. Data on 1, 3, and 7-day mortality was now added to Table 2. In addition, we now provide data on 1-year mortality and a survival analysis. Page 9, lines 212-215, new Figure 2.

 Comment #2.2: The number of patients analyzed appears to be 281, but in the first sentence of the Discussion, 282 patients are presented. I encourage you to provide an accurate number.

Response to comment #2.2: Thank you very much for omitting this mistake. This was now corrected and the number of 281 patients is now indicated homogenously throughout the manuscript. Page 11, line 233.

Comment #2.3: If possible, it would be meaningful to describe whether there are differences in adverse events depending on age groups and gender.

Response to comment #2.3: In our patient population, only 71 (25.3%) women were present. Given the low proportion of women represented in our cohort, conclusive interpretation of sex-specific data is limited. We now provide additional data on adverse events, stratified by 1) sex, and 2) age categories as  Supplementary Table 1.1 and Table 1.2.
